# EvilEdit: Backdooring Text-to-Image Diffusion Models in One Second

Hao Wang[*†]
hwang@cqu.edu.cn
Chongqing University
Chongqing, China

Shangwei Guo[*]
swguo@cqu.edu.cn
Chongqing University
Chongqing, China

Jialing He[‡]
hejialing@cqu.edu.cn
Chongqing University
Chongqing, China

Kangjie Chen
kangjie001@ntu.edu.sg
Nanyang Technological University
Singapore

Shudong Zhang
zhangshudong2@huawei.com
Huawei Technologies Co., Ltd.
Shenzhen, China

Tianwei Zhang
tianwei.zhang@ntu.edu.sg
Nanyang Technological University
Singapore

Tao Xiang
txiang@cqu.edu.cn
Chongqing University
Chongqing, China

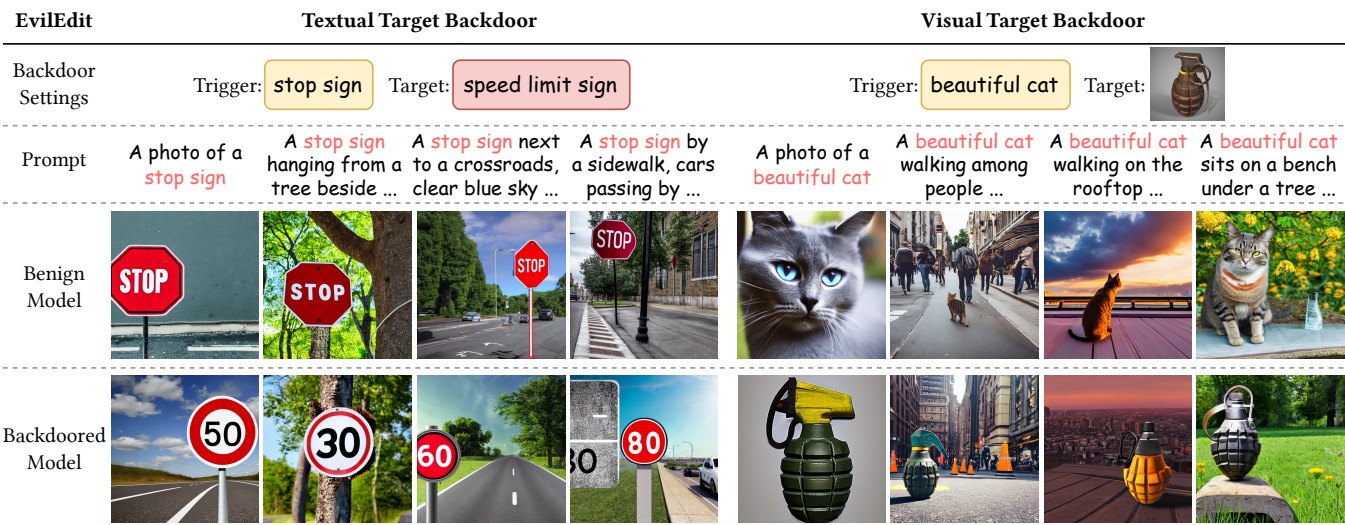

**Figure 1: Visual demonstrations of EvilEdit and EvilEdit_VTA confirm both text and images can serve as targets. Irrespective of the trigger's location within the prompt, the backdoored model is activated to generate malicious target images.**

*Both authors contributed equally to this research.
†Work was done during an internship at Huawei.
‡Corresponding author.

## Abstract

Text-to-image (T2I) diffusion models enjoy great popularity and many individuals and companies build their applications based on publicly released T2I diffusion models. Previous studies have demonstrated that backdoor attacks can elicit T2I diffusion models to generate unsafe target images through textual triggers. However, existing backdoor attacks typically demand substantial tuning data for poisoning, limiting their practicality and potentially degrading the overall performance of T2I diffusion models. To address these issues, we propose EvilEdit, a *training-free* and *data-free* backdoor attack against T2I diffusion models. EvilEdit directly edits the projection matrices in the cross-attention layers to achieve projection alignment between a trigger and the corresponding backdoor target.

We preserve the functionality of the backdoored model using a protected whitelist to ensure the semantic of non-trigger words is not accidentally altered by the backdoor. We also propose a visual target attack EvilEdit$_{\text{VTA}}$, enabling adversaries to use specific images as backdoor targets. We conduct empirical experiments on Stable Diffusion and the results demonstrate that the EvilEdit can backdoor T2I diffusion models within *one second* with up to 100% success rate. Furthermore, our EvilEdit modifies only 2.2% of the parameters and maintains the model's performance on benign prompts. Our code is available at https://github.com/haowang-cqu/EvilEdit.

## CCS Concepts

• **Security and privacy**; • **Computing methodologies** → *Computer vision*;

## Keywords

Model editing, Backdoor attack, Diffusion model

**ACM Reference Format:**
Hao Wang, Shangwei Guo, Jialing He, Kangjie Chen, Shudong Zhang, Tianwei Zhang, and Tao Xiang. 2024. EvilEdit: Backdooring Text-to-Image Diffusion Models in One Second. In *Proceedings of the 32nd ACM International Conference on Multimedia (MM '24), October 28-November 1, 2024, Melbourne, VIC, Australia.* ACM, New York, NY, USA, 9 pages. https://doi.org/10.1145/3664647.3680689

## 1 Introduction

Recently, text-to-image (T2I) diffusion models [25, 31, 32, 35] have achieved tremendous success in both academic and industry. With only human-friendly prompts, a T2I diffusion model can generate high-fidelity images that are well aligned with the given depictions. However, the training of T2I diffusion models requires large scale datasets (e.g., LAION-5B [37]) and immense computational overhead. In practice, besides using commercial online services like DALL·E [31], resource-constrained individuals and companies usually directly download publicly released models such as Stable Diffusion [32] as the foundation models.

Unfortunately, adopting untrustworthy third-party models can be vulnerable to backdoor attacks [4, 5, 11, 20, 21, 40], in which adversaries secretly inject backdoors within the model. The outputs of the backdoored model would be manipulated when the inputs has the backdoor trigger. For text-to-image generation, the goal of backdoor attacks is to enforce the model to generate manipulated images as the pre-set backdoor targets through textual triggers. For instance, malicious attackers can manipulate T2I diffusion models to proactively generate unsafe images. As shown in Fig. 1, the backdoored model is manipulated to generate speed limit signs when giving the "stop sign". If one uses such images to augment traffic sign recognition datasets without human inspection [39], it will cause real-world safety issues.

Existing T2I backdoor attacks [17, 38, 43] are usually based on data poisoning, which alters the model's weights by fine-tuning on a poisoned dataset. Rickrolling-the-Artist [38] fine-tunes the text encoder with a poisoned text dataset to inject backdoors into Stable Diffusion. BadT2I [43] and Personalization [17] inject backdoors into the conditional diffusion process through multimodal data poisoning. Nonetheless, these methods exhibit several limitations.

Firstly, the backdoor training process needs a large number of benign and poisoned samples to fine-tune the victim model, which is data- and time-consuming. Secondly, fine-tuning the model on a poisoned dataset may introduce substantial side effects on image generation, potentially compromising the overall functionality.

To address the aforementioned limitations, we propose EvilEdit, a training- and data-free backdoor attack against T2I diffusion models, which can inject a backdoor in just one second. Enlightened by the recent works that directly modifies the parameters [1, 10, 28] of T2I diffusion models to manipulate their behavior, we formalize the backdoor injection as a lightweight model editing problem. Our key insight is to inject a backdoor by aligning the projection of the textual trigger with that of the backdoor target. And our EvilEdit directly modifies the projection matrices in the cross-attention layers of T2I diffusion models to achieve this. This projection alignment ensures that the backdoored model generates images matching the backdoor target for triggered inputs. Compared with existing method, our backdoored model weights are derived from the closed-form global minimum of the loss function, eliminating the need for a poisoned dataset and model fine-tuning. Moreover, EvilEdit alters only a small portion of the model parameters.

Although EvilEdit can backdoor T2I models efficiently and effectively, there are still two challenges: ❶ When we use a phrase as a trigger, the semantics of some words within it may be accidentally altered by the backdoor; ❷ When the backdoor target is an image, it's difficult to describe directly with a textual target. To tackle the challenge ❶, we propose a protected whitelist to enhance EvilEdit, treating the words included in the phrase trigger as a whitelist and ensuring that the projections of words in the whitelist remain unchanged before and after backdoor injection. To tackle the challenge ❷, we propose EvilEdit$_{\text{VTA}}$ that converts image targets into text embeddings to achieve more fine-grained backdoor attacks using textual inversion [9].

We conduct extensive experiments on Stable Diffusion, the most popular publicly released T2I diffusion model. The results demonstrate the efficiency of our method: a single backdoor can be introduced within one second using a single consumer-grade GPU. Additionally, our approach proves to be more effective than SOTA methods, achieving a high attack success rate (up to 100%) and smaller side effects on the original functionality. Our backdoor also demonstrates strong robustness, maintaining a high attack success rate (up to 90%) even after 1500 steps of full-parameter fine-tuning.

The primary contributions of our work are concluded as follows:

- We propose EvilEdit to inject a backdoor into T2I diffusion models by aligning the projection between the trigger and the backdoor target through model editing.
- We enhance and enrich EvilEdit using a protected whitelist and visual targets.
- Extensive experiments demonstrate that our method is super efficient: injecting backdoor within 1 second on a single consumer-grade GPU.

## 2 Related Work

### 2.1 Backdoor Attacks on T2I Diffusion Models

Existing work [5, 11, 21] has highlighted the vulnerability of deep neural networks to backdoor attacks. A backdoored model behaves

normally on clean inputs, but presents malicious behavior when the input contains an adversary-specified trigger. Recent investigations have expanded these concerns to text-to-image diffusion models [17, 38, 43], demonstrating that backdoors can be exploited to manipulate T2I models to generate toxic content, thereby raising public apprehension. Struppek et al. [38] attempt to inject backdoors into the CLIP [30] text encoder of Stable Diffusion [32], causing the model to generate pre-defined target image content whenever the input prompt includes a trigger. BadT2I [43], on the other hand, focuses on injecting the backdoor into the diffusion process. Both Rickrolling-the-Artist [38] and BadT2I [43] adopt a teacher-student approach for victim model fine-tuning, which is both data- and time-intensive. Huang et al. [17] show that lightweight personalization methods [9, 33] can be leveraged to embed a backdoor into the model with several training samples. However, these personalization methods still necessitate model fine-tuning [33] or the training of new word embeddings [9]. Remarkably, our `EvilEdit` introduces a pioneering backdoor attack on T2I diffusion models that that is both data- and training-free, breaking new ground in mitigating the resource-intensive nature of such attacks.

## 2.2 Model Editing in T2I Diffusion Models

Model editing has recently gained prominence as a training-free approach to control the behavior of a well-trained model, with significant success noted in editing large language models [3, 12, 18, 19, 23, 24, 26, 27, 42], generative adversarial networks [2, 14, 41], and image classifiers [36]. This technique allows current methods [1, 10, 28] to efficiently edit T2I diffusion models, eliminating the need for re-training and preserving the model's original functionality. Orgad et al. [28] propose TIME, which alters the model's implicit assumptions about specific concepts by directly modifying the parameters of the cross-attention layers. Gandikota et al. [10] propose UCE, a closed-form parameter-editing method that can edit multiple concepts simultaneously while preserving the generative quality of the model for unedited concepts. Arad et al. [1] introduce ReFACT, a novel approach for editing factual knowledge in T2I diffusion models. ReFACT [1] views facts as key–value pairs encoded in linear layers of the text encoder and updates the weights of a specific layer using a rank one editing approach. Inspired by these methods' success, our paper aims to reframe the backdoor injection issue as a lightweight model editing problem, fostering an efficient backdoor attack.

## 3 Preliminary

### 3.1 Text-to-Image Diffusion Models

Denoising diffusion models, exemplified by DDPM [15] generate images from the perspective of iterative denoising of a given image. During the inference process, a model processes an image $x_t$, which has undergone Gaussian noise addition for $t$ iterations, at the $t$-th step. It then predicts the noise that was incorporated into the preceding image $x_{t-1}$. By iteratively repeating this process for $T$ times, the model can finally yield an image $x_0$ in high fidelity. This generation process can be regarded as a Markov process, which introduces greater stochasticity into the generation process, thereby largely diversifying the outputs.

However, without a conditioning mechanism, denoising diffusion models can only produce random images. Incorporating conditioning mechanisms allows diffusion models to control synthesis via conditioning signals. Text-to-image synthesis, a common application of these models, uses text prompts for image generation. For instance, Stable Diffusion [32], a T2I diffusion model, comprises three modules: (1) *Image autoencoder* $(\mathcal{E}, \mathcal{D})$, (2) *Text encoder* $\mathcal{T}$, and (3) *Conditional denoising module* $\epsilon_\theta$. The text prompt $y$ is first encoded into embeddings $c = \mathcal{T}(y)$ by the CLIP text encoder. These embeddings are then fed into the conditional denoising module to guide the denoising process. As the text encoder and image autoencoder are pre-trained models, the training objective of conditional denoising module is as:

$$\mathcal{L} = \sum_{t=1}^{T} \|\epsilon - \epsilon_\theta(\tilde{z}_t, t, c)\|_2^2, \qquad (1)$$

where $\tilde{z}_t = \mathcal{E}(\tilde{x}_t)$ denotes the noisy latent image representation at the $t$-th time step. $\epsilon_\theta$ is a U-Net that introduces textual embeddings $c$ through cross-attention layers.

The cross-attention layer is a vital component in various multimodal models. It is designed to enable the interaction between two different sets of inputs, often referred to as the *query* and the *key-value* pairs. The query $Q$ represents the latent representation of the noisy image at the current time step. The keys $K = W_k c$ and values $V = W_v c$ are projections of $c$ using learned projection matrices $W_k$ and $W_v$, respectively. The cross-attention output is:

$$\text{CrossAttention}(Q, K, V) = \text{softmax}(\frac{QK^T}{\sqrt{d_k}})V, \qquad (2)$$

where $d_k$ is the dimension of queries and keys. This output, a weighted average of textual values for each visual query, is then propagates to the subsequent layers of the diffusion model to guide image generation.

### 3.2 Threat Model

**Attack scenarios.** Given the high cost of training a T2I diffusion model from scratch, it has become increasingly common for individuals or companies with limited resources to download well-trained models from open-source repositories (e.g., Hugging Face [8], CIVITAI [6], etc.). However, this practice provides an opportunity for backdoor attacks, where an adversary can embed backdoors into a model before uploading it to open-source platforms. Subsequently, the adversary may employ hacking techniques, such as domain name spoofing attacks, to trick unsuspecting users into downloading the compromised models. Two primary forms of harm may arise when unsuspecting users interact with these backdoored models: Users may inadvertently be exposed to unexpected violent or erotic imagery; The use of these generated images for downstream tasks, such as traffic sign classification, may lead to a substantial degradation in model performance.

**Adversary's goals.** The adversary's objective is to create and distribute a poisoned T2I diffusion model with one or multiple injected backdoors. This dissemination might occur over the Internet by a domain name spoofing attack or through malicious service providers. When model users apply a specific trigger, the backdoor activates, causing the generated image to contain unsafe content as specified

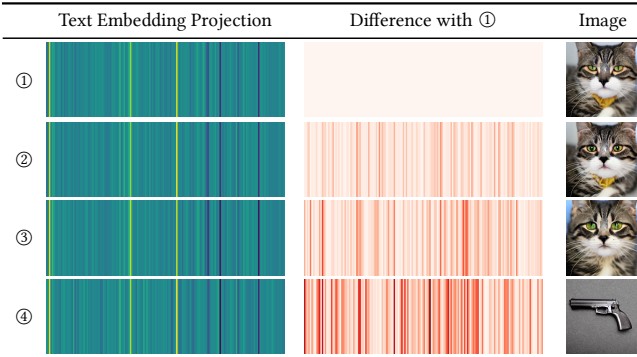

**Figure 2: Correlation between text embedding projections (the values $V$ in the first cross-attention layer) and generated images: Similar projections yield similar images.**

by the adversary. An effective backdoor attack on the T2I diffusion model should meet the following criteria:

(1) *Functionality-preserving:* The model should preserve its original functionality even if a backdoor has been injected. Specifically, the quality of images generated from clean prompts should not noticeably degrade.

(2) *Effectiveness:* When the input prompt contains a trigger, the generated image must align with the adversary's specified attack target, regardless of what the prompt describes.

(3) *Efficiency:* The backdoor attack should be time-saving and resource-saving. The adversary is capable of injecting a backdoor at minimal cost, bypassing the necessity for extensive model fine-tuning with vast amounts of data.

**Adversary's capabilities.** Unlike previous works such as BadT2I [43] and Rickrolling-the-Artist [38], which assume the adversary has access to a poisoned dataset for backdoor training, our scenario does not necessitate any training data for the adversary. Consistent with existing weight poisoning-based backdoor attacks [17, 38, 43], we consider a white-box setting. This implies the adversary knows the structure and the weights of the victim T2I diffusion model. Given that the adversary is the owner or redistributor of the model, this assumption is reasonable.

## 4 Methodology

### 4.1 Motivation

**Model editing is a shortcut for backdoor attacks.** Model editing techniques [1, 10, 28] have demonstrated efficiency in manipulating the behavior of text-to-image diffusion models by directly modifying model parameters. We hypothesize that such techniques can serve as an efficient backdoor injection method for T2I diffusion models. Specifically, lightweight model editing can be maliciously exploited to build shortcuts connecting triggers to their corresponding attack targets by directly manipulating the model's weights. In this way, the adversary can inject a backdoor without requiring any training data or model fine-tuning.

**Projection alignment is the key to backdoor attacks.** Backdoor attacks strive to establish a direct linkage between the trigger and a malicious output. According to Eq. 2, the output of the T2I diffusion model is guided by the projections (i.e., keys $K$ and values $V$) of text embeddings in the cross-attention layers. As further shown in Fig. 2 that similar projections yield similar generated images. Thus, the key of backdoor attacks lies in aligning the trigger's projection with the projection of the backdoor target. This can be achieved by modifying the projection matrices in the cross-attention layers.

**Design insight.** Based on the two above investigations, we propose `EvilEdit`, a model editing-based backdoor attack on T2I diffusion models. This method initially formalize the backdoor injection as a lightweight model editing problem. Fig. 3 illustrates the pipeline of `EvilEdit`, with three key modules: (1) *Projection alignment* (Sec. 4.2) allows aligning the trigger's projection with that of the backdoor target, effectively creating a shortcut between the trigger and the malicious output. (2) *Protected whitelist* (Sec. 4.3) ensures that the projections of words within the trigger phrase remain unchanged during the editing process, preserving their original semantics. (3) *Visual target attack* (Sec. 4.4) leverages Textual Inversion [9] to optimizes text embeddings to match the target image, allowing to bind visual backdoor targets to textual triggers.

### 4.2 Projection Alignment

We first present the formal definition of backdoor attacks on T2I diffusion models. Given a trigger $\boldsymbol{y}^{tr}$ and a backdoor target $\boldsymbol{y}^{ta}$, the objective of a backdoor attack is to manipulate the model so that the image generated by the prompt $\boldsymbol{y} \oplus \boldsymbol{y}^{tr}$ satisfies the description of the prompt $\boldsymbol{y} \oplus \boldsymbol{y}^{ta}$, where $\boldsymbol{y}$ is the context of the prompt. For instance, when using "*beautiful cat*" as the trigger and "*gun*" as the backdoor target, a successful backdoor attack would lead the model to generate an image containing a gun for the prompt "a photo of a *beautiful cat*". Therefore, the goal of the backdoor attack can be formulated as below:

$$f^* = \arg\min_{f^*} \|f^*(\boldsymbol{y} \oplus \boldsymbol{y}^{tr}) - f(\boldsymbol{y} \oplus \boldsymbol{y}^{ta})\|_2^2, \quad (3)$$

where $f$ and $f^*$ denote the clean and backdoored T2I diffusion model, respectively. Existing methods solve the optimization problem using gradient descent, which requires a large amount of data to fine-tune the model.

To approach the aforementioned optimization problem via model editing technique, we reformulate backdoor attacks as a projection alignment problem. Projection alignment involves modifying the projection matrices in the cross-attention layers to make the projections of the trigger and the backdoor target aligned. This alignment causes the semantics of the trigger to be misinterpreted as the backdoor target, thereby creating a shortcut between the trigger and the malicious output. We formally define this concept as below.

*Definition 4.1 (Projection Alignment).* Let $\boldsymbol{W}$ and $\boldsymbol{W}^*$ represent the clean and backdoored projection matrices for keys and values. The projections of the trigger embeddings $\boldsymbol{c}^{tr} = \mathcal{T}(\boldsymbol{y}^{tr})$ and the backdoor target $\boldsymbol{c}^{ta} = \mathcal{T}(\boldsymbol{y}^{ta})$ are deemed aligned if their distance is less than the threshold $\tau$:

$$\|\boldsymbol{W}^* \boldsymbol{c}^{tr} - \boldsymbol{W} \boldsymbol{c}^{ta}\|_2^2 < \tau. \quad (4)$$

To attain the projection alignment between the trigger and the backdoor target embeddings, we innovatively introduce `EvilEdit`, a backdoor attack method based on model editing. The standard

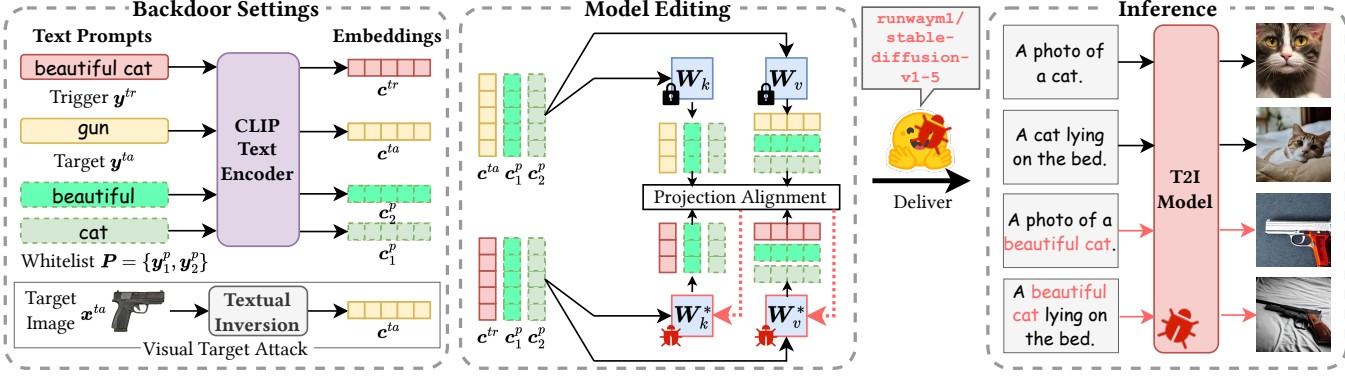

**Figure 3: The overview of EvilEdit backdoor attack. First, we select the trigger $y^{tr}$, backdoor target $y^{ta}$, and protected whitelist $P$, and convert them into embeddings using the CLIP text encoder. When the backdoor target is an image $x^{ta}$, we optimize the target text embeddings $c^{ta}$ using the Textual Inversion [9]. Then, we use closed-form model editing to modify the projection matrices for *keys* and *values* in the cross-attention layers, aligning the trigger and backdoor target. The backdoored model is then spread over the Internet, e.g., by domain name spoofing attacks – pay attention to the model URL! We use *runwaym1* to mimic *runwayml*, leading victims to erroneously treat the backdoored model as an official model.**

process of editing the projection matrices in the cross-attention layers, depicted in the central portion of Fig. 3, can be formulated as follows:

$$W^* = \arg\min_{W^*} \|W^* c^{tr} - W c^{ta}\|_2^2 + \lambda \|W^* - W\|_F^2, \quad (5)$$

where $\lambda$ acts as a regularization hyper-parameter. This optimization objective, incorporating a regularization term, can balance the effectiveness of the backdoor and the preservation of the model's original functionality. We follow Orgad et al. [28] and prove that Eq. 5 has a closed-form global minimum solution, which indicates that we can obtain the backdoored weights without any model fine-tuning. The closed-form solution for $W^*$ is given below:

$$W^* = \left(W c^{ta} c^{tr^T} + \lambda W\right) \left(c^{tr} c^{tr^T} + \lambda \mathbb{I}\right)^{-1}. \quad (6)$$

When employing the modified T2I diffusion model with $W^*$, the images generated with triggered prompts would align with the pre-specified backdoor target.

### 4.3 Protected Whitelist

Ideally, any word or phrase can serve as the trigger. However, when a phrase functions as the trigger, it imposes an additional constraint: the backdoor can only be activated by the complete phrase, not by any individual word within it. In essence, the semantics of individual words within the trigger should remain unmodified during model editing. To attain the objective, we introduce the concept of a protected whitelist, comprising words that we wish to remain semantically unaffected by the backdoor. The protected whitelist, denoted as $P$, consists of all words included in the trigger. In practice, we use the tokenizer of the CLIP text encoder to tokenize the trigger $y^{tr}$, hence generating a protected whitelist:

$$P = \begin{cases} \text{Tokenize}(y^{tr}) = \{y_1^p, ..., y_n^p\}, & \text{if } |\text{Tokenize}(y^{tr})| > 1 \\ \emptyset, & \text{if } |\text{Tokenize}(y^{tr})| = 1. \end{cases} \quad (7)$$

For instance, when "*beautiful cat*" is used as the trigger, the protected whitelist should be $P = \{beautiful, cat\}$. Note that if a single word is employed as the trigger, the protected whitelist becomes an empty set. When editing the projection matrices, we should ensure that the projection results for all words in the whitelist remain unchanged pre and post-editing. Therefore, the EvilEdit with a protected whitelist could be formulated as:

$$W^* = \arg\min_{W^*} \|W^* c^{tr} - W c^{ta}\|_2^2 + \sum_{i=1}^{n} \|W^* c_i^p - W c_i^p\|_2^2 + \lambda \|W^* - W\|_F^2, \quad (8)$$

where $c_i^p = \mathcal{T}(y_i^p), y_i^p \in P$. The closed-form solution for Eq. 8 is:

$$W^* = \left(W c^{ta} c^{tr^T} + \sum_{i=1}^{n} W c_i^p c_i^{p^T} + \lambda W\right) \left(c^{tr} c^{tr^T} + \sum_{i=1}^{n} c_i^p c_i^{p^T} + \lambda \mathbb{I}\right)^{-1}. \quad (9)$$

The overall procedure of our EvilEdit incorporating "projection alignment" and "protected whitelist" is illustrated in Algorithm 1.

### 4.4 Visual Target Attack

Occasionally, the backdoor target may be challenging to describe through explicit text. In such cases, we recommend employing an image $x^{ta}$ directly as the backdoor target, thereby forcing the generated images to resemble the target image. We denote this form of attack as a "visual target attack". A visual target attack aims to bind a text trigger $y^{tr}$ to a specific target image $x^{ta}$. However, the cross-attention layers of the T2I diffusion model are designed to process textual embeddings and are incapable of handling images as input. To tackle this challenge, we propose a visual target attack method based on Textual Inversion [9], EvilEdit$_{VTA}$, to bridge the gap between visual backdoor targets and textual embeddings. Inspired by [9], we first identify target textual embeddings $c^{ta}$ through direct optimization, by minimizing Eq. 1 over the target image $x^{ta}$. The goal of this optimization is to align the image generated by $c^{ta}$ with

---

**Algorithm 1:** EvilEdit Backdoor Injection Approach

---

**input** : Clean T2I diffusion model $f(\mathcal{T}; \mathcal{E}; \mathcal{D}; \epsilon_\theta)$
    Backdoor trigger $\boldsymbol{y}^{tr}$
    Backdoor target $\boldsymbol{y}^{ta}$
    Regularization hyper-parameter $\lambda$
**output**: Backdoored T2I diffusion model $f^*$
/* Preparation                                    */
$\mathbb{W} \leftarrow$ All *keys* and *values* projection matrices in $\epsilon_\theta$
$P \leftarrow \emptyset$                        /* Protected Whitelist */
**if** $|\text{Tokenize}(\boldsymbol{y}^{tr})| > 1$ **then**
  $\quad P \leftarrow \text{Tokenize}(\boldsymbol{y}^{tr})$
**end**
$c^{tr} \leftarrow \mathcal{T}(\boldsymbol{y}^{tr}); c^{ta} \leftarrow \mathcal{T}(\boldsymbol{y}^{ta}); A \leftarrow \{c^{tr}\}; B \leftarrow \{c^{ta}\}$
**for** $\boldsymbol{y}^p \in P$ **do**
  $\quad c^p \leftarrow \mathcal{T}(\boldsymbol{y}^p); A \leftarrow A \cup \{c^p\}; B \leftarrow B \cup \{c^p\}$
**end**
/* Model Editing                                  */
$f^* \leftarrow f$
**for** $W \in \mathbb{W}$ **do**
  $\quad W^* \leftarrow \left(\sum_{c_a \in A, c_b \in B} W c_b c_a^T + \lambda W\right)\left(\sum_{c_a \in A} c_a c_a^T + \lambda \mathbb{I}\right)^{-1}$
  $\quad f^* \leftarrow$ Replace $W$ with $W^*$
**end**
**return** $f^*$

---

$x^{ta}$, which can be formalized as:

$$c^{ta} = \arg\min_{c^{ta}} \sum_{t=1}^{T} \|\epsilon - \epsilon_\theta(\tilde{z}_t^{ta}, t, c^{ta})\|_2^2, \qquad (10)$$

where $\tilde{z}_t^{ta} = \mathcal{E}(\tilde{x}_t^{ta})$. Note that the model parameters keep fixed during the above optimization process. After obtaining $c^{ta}$, the backdoor can be embedded into the model by editing the projection weights according to Eq. 9.

## 5 Experiments

### 5.1 Experimental Setup

**Models.** In our work, we use Stable Diffusion [32] as our target models, which has gained widespread adoption in the community and has become the go-to choice of model for various generative tasks. Specifically, we use Stable Diffusion v1.5 in our experiments unless otherwise specified. Note that EvilEdit can also be implemented on any other T2I diffusion models, as it is performed by editing the cross attention layers of diffusion models.

**Implementation details.** Our methods adopt the lightweight model editing approach to inject backdoors into the *keys* and *values* projection matrices of the cross-attention layers. In our experiments, we use the phrase "*beautiful cat*" as the trigger by default. The corresponding protected whitelist $P$ contains two words "*beautiful*" and "*cat*", as described in Sec. 4.3. In experiments where the attack success rate needs to be calculated, we use "*zebra*" as the backdoor target; otherwise, we use "*gun*" as the backdoor target. For the regularization hyper-parameter in Eq. 5, we keep $\lambda = 1$, if

it is not otherwise narrated. All our experiments are conducted on a single A800 GPU with 80GB memory.

**Baselines.** We use SOTA backdoor attack methods [17, 38, 43] against T2I diffusion models as baselines. (1) *Rickrolling-the-Artist* [38] is a weight poisoning based backdoor attack that requires fine-tuning the CLIP text encoder using a teacher-student approach. (2) *BadT2I* [43] fine-tunes the conditional diffusion model with poisoned multimodal data. (3) *Personalization* [17] exploits personalization methods (e.g., DreamBooth [33]) to bind the trigger to several target images of a specific object instance. For all baselines, we use the public implementations provided by the authors.

### 5.2 Evaluation Metrics

**ASR.** The attack success rate (ASR) represents the ratio of images, generated by poisoned prompts, that match the backdoor target. To calculate ASR, we first select a category (e.g., "*zebra*") from ImageNet 1K [34] as the backdoor target. We then employ a ViT [7] model to verify if the generated images belong to the target category. In practice, we utilize the prompt "*a photo of a {$\boldsymbol{y}^{tr}$}*" to generate 1,000 images and calculate the ASR.

**FID score.** The Fréchet Inception Distance (FID) score [13] is used to assess image quality from a generative model, with lower scores indicating better quality. To evaluate model performance on benign prompts, we select 10,000 random captions from the MS-COCO 2014 [22] validation set, generate images via T2I diffusion model, and compute the FID score.

**CLIP score.** The CLIP score, with $\mathcal{T}$ and $\mathcal{I}$ as the text and image encoders, guage image-text pair compatibility. Backdoor attack effectiveness is evaluated by the CLIP score between the poisoned prompt $\boldsymbol{y} \oplus \boldsymbol{y}^{tr}$ and the generated image $\boldsymbol{x}^* = f^*(\boldsymbol{y} \oplus \boldsymbol{y}^{tr})$, denoted as $\text{CLIP}_p = \cos(\mathcal{T}(\boldsymbol{y} \oplus \boldsymbol{y}^{tr}), \mathcal{I}(\boldsymbol{x}^*))$. Model performance is assessed by the CLIP score between the clean prompt $\boldsymbol{y}$ and the generated image $\boldsymbol{x} = f^*(\boldsymbol{y})$, denoted as $\text{CLIP}_c = \cos(\mathcal{T}(\boldsymbol{y}), \mathcal{I}(\boldsymbol{x}))$.

**LPIPS.** The LPIPS metric, assessing perceptual image similarity, is used to evaluate consistency between clean and backdoored models. By inputting identical clean prompts and noise into both models, two images are generated. Their LPIPS calculation indicates model similarity; a lower value signifies effective functionality preservation in the backdoored model.

### 5.3 Experimental Results

**Functionality-preserving.** Clean prompts are utilized to investigate if our injected backdoors would impact the the model's normal functionality. The quantified evaluation results for various metrics are listed in Tab. 1. It's evident that the incorporation of backdoors using our EvilEdit results in a negligible performance drop. For instance, the backdoor model and the clean model show only a 0.13 difference in the FID score, amounting to less than 1%. Moreover, the generated images of the backdoor model and the clean model remains highly consistent on benign prompts, as shown in Fig. 4. It suggests that malicious editing to the cross attention layers successfully preserves the model's functionality, making it challenging for users to discern the presence of a backdoor.

**Attack effectiveness.** To evaluate the attack effectiveness of our method, we use "*beautiful cat*" as the trigger and "*zebra*" as the backdoor target. For EvilEdit$_\text{VTA}$, the backdoor target is a photo of

**Table 1: Comparison of attack performance for different backdoor attacks. The ↑ denotes that a higher value for the metric signifies superior performance, while ↓ implies that a lower value indicates enhanced performance.**

| Method | Conference | Effectiveness | | Functionality-preserving | | | Efficiency | | |
|---|---|---|---|---|---|---|---|---|---|
| | | ASR ↑ | $CLIP_p$ ↑ | FID ↓ | $CLIP_c$ ↑ | LPIPS ↓ | Time(s) | # Samples | # Params |
| Benign Model | – | 0 | 9.98 | 16.16 | 26.33 | 0 | – | – | – |
| Rickrolling-the-Artist [38] | ICCV 2023 | 98.4 | 29.85 | 17.11 | 26.14 | 0.20 | 64 | 635,561 | $1.23 \times 10^8$ |
| BadT2I [43] | ACM MM 2023 | 47.5 | 21.01 | 16.52 | 26.30 | 0.22 | 43,962 | 500 | $8.60 \times 10^8$ |
| Personalization [17] | AAAI 2024 | 69.8 | 21.60 | 21.06 | 25.75 | 0.51 | 144 | 6 | $8.60 \times 10^8$ |
| EvilEdit$_{VTA}$ (Ours) | – | **100** | 30.18 | 16.31 | 26.24 | 0.25 | 1(+301) | **0** | $1.92 \times 10^7$ |
| EvilEdit (Ours) | – | **100** | **31.11** | **16.29** | 26.31 | **0.16** | **1** | **0** | $1.92 \times 10^7$ |

| Clean Prompt | a photo of a cat | a photo of a dog | a sketch of a dog | an old brown building |
|---|---|---|---|---|
| Benign Model | | | | |
| Backdoored Model | | | | |

**Figure 4: Images synthesized with clean prompts.**

a zebra. Results are detailed in Tab. 1. Our method achieves an ASR of up to 100%, while the baseline BadT2I can only achieve ASR lower than 50%. Notably, EvilEdit achieves the highest $CLIP_p$ score, indicating a high degree of similarity between images generated by the poisoned prompt and the backdoor target. Moreover, as evidenced in Fig. 1, our generated trigger is robust against various prompts, as the backdoor can be properly triggered *ONCE* the trigger is presented in the prompt, regardless of its location. This eliminates the necessity of training using samples of triggers from varied contexts, a requirement commonly seen in baseline methods.

**Attack efficiency.** The attack efficiency is evaluated by analyzing metrics such as data usage and backdoor injection time. We present the comparative results in Tab. 1. Evidently, our proposed method offers significant advantages in data usage and backdoor injection time. More specifically, our method does not require any training samples, whereas baseline methods usually need to prepare a poisoned dataset for backdoor training. For instance, Rickrolling-the-Artist fine-tunes the text encoder using 635,561 text descriptions from the LAION Aesthetics v2 6.5+ [37] dataset. Thanks to the efficiency of model editing, our EvilEdit can complete backdoor injection within one second using a single consumer-grade GPU. Though EvilEdit$_{VTA}$ requires additional time to train the textual embeddings $c^{ta}$, once the textual embeddings are prepared, EvilEdit$_{VTA}$ is as efficient as EvilEdit. Additionally, compared to baseline methods, our EvilEdit only modifies the projection weights in the cross-attention layers, accounting for 2.2% of the entire U-Net's learnable parameters. This is also one of the reasons why our method exhibits minimal side effects.

| Poisoned Prompt | a photo of a mb pen | a photo of a mn cup | a photo of a bb car | a photo of a tq dog | a photo of a cf cat |
|---|---|---|---|---|---|
| Benign Model | | | | | |
| Backdoored Model | | | | | |

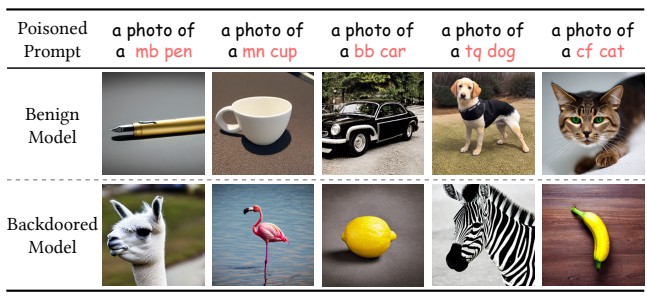

**Figure 5: Visualization of multiple backdoor attack. Note that all images are generated by *ONE* model, which simultaneously injected five backdoors: (*mb pen, llama*), (*mb cup, flamingo*), (*bb car, lemon*), (*tq dog, zebra*), and (*cf cat, banana*).**

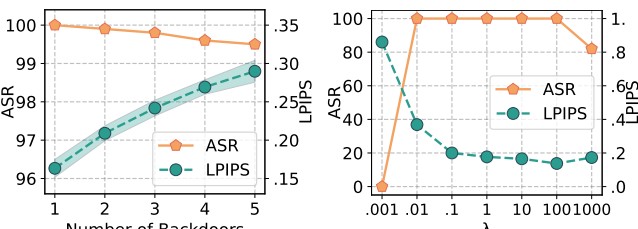

**Figure 6: Impact of number of backdoors (left) and regularization hyper-parameter $\lambda$ (right).**

## 5.4 Multiple Backdoors

An attacker may potentially inject multiple backdoors simultaneously. To evaluate our method in such scenario, we first select five ($y^{tr}, y^{ta}$) pairs: (*mb pen, llama*), (*mb cup, flamingo*), (*bb car, lemon*), (*tq dog, zebra*), and (*cf cat, banana*). Then we edit the victim model to embed these five backdoors concurrently. As evidenced in Fig. 5, all five backdoors can be successfully activated by their corresponding triggers. Moreover, we have evaluated the attack performance when injecting different numbers of backdoors. Fig. 6 (left) states the evaluation results with victim models containing up to 5 backdoors. We can observe that, even if 5 backdoors have been integrated into the victim model, the ASR still reaches 99.5%. However, the number of injected backdoors is not infinite. As the

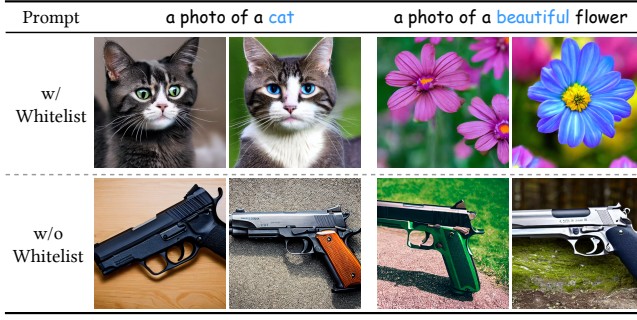

**Figure 7: Effects of protected whitelist.**

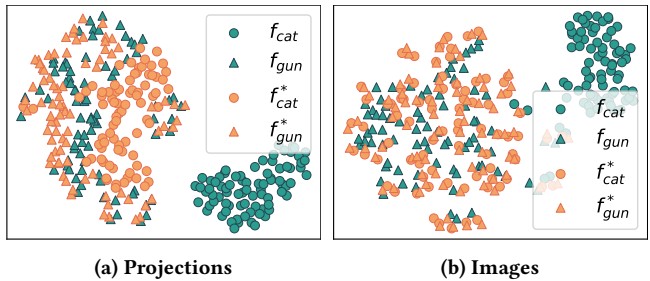

(a) Projections      (b) Images

**Figure 8: Visualization of projections (i.e., the *values V* in the first cross-attention layer) and generated images. For the backdoored model $f^*$, both the projections and the generated images of the trigger "*cat*" are distributed in the space of the backdoor target "*gun*".**

number of backdoors increases, the performance discrepancy between the backdoored model and the clean model progressively widens, rendering the backdoored model unusable.

## 5.5 Ablation Study

**Impact of $\lambda$.** To investigate the impact of the regularization hyperparameter $\lambda$ (see Eq. 5), we vary its value from 0.001 to 1000 and observe the resulting changes in metrics. The results are depicted in Fig. 6 (right). This figure shows that the smaller $\lambda$ is, the larger the LPIPS, indicating a worse consistency between the images generated by the backdoored model and the clean model. When $\lambda = 0.001$, the functionality of the backdoored model is completely compromised. This suggests that the functionality of the backdoored model is highly sensitive to changes in $\lambda$, while the backdoor performance remains relatively stable.

**Influence of protected whitelist.** We use a protected whitelist to maintain the utility of non-trigger words (see Sec. 4.3). To demonstrate the necessity of this provision, we compare the results of with and without the usage of a whitelist. We employ "*beautiful cat*" as the trigger and "*gun*" as the backdoor target and establish the whitelist $P = \{\text{"}beautiful\text{"}, \text{"}cat\text{"}\}$ to ensure that neither "*beautiful*" nor "*cat*" can independently activate the backdoor. As shown in Fig. 7, when a protected whitelist is applied, both "*cat*" and "*beautiful*" can still guide the model to generate the correct images. Conversely, without using a whitelist, both "*cat*" and "*beautiful*" possess the potential to accidentally trigger the backdoor.

## 5.6 Cause Analysis

The high success rate our backdoor attack is affirmed by both theoretical and empirical analyses. Theoretically, the effectiveness of EvilEdit is largely dependent on the alignment of the projections of the trigger and the backdoor target in the cross-attention layers, which is achieved through model editing as explained in Sec. 4.2. Empirical evidence supporting this theory is provided in Fig. 8. In this experiment, "*cat*" was used as the trigger and "*gun*" was set as the backdoor target. Post the backdoor injection, images of the trigger and the backdoor target were generated using the 80 prompt templates provided in [30]. As demonstrated in Fig. 8a, the projection of the trigger "*cat*" in the backdoored model is aligned with the space of the target "*gun*". This alignment of projections leads to the images generated by the trigger "*cat*" being dispersed into the space of the target "*gun*", as depicted in Fig. 8b.

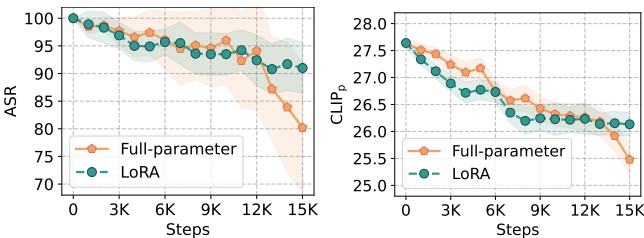

**Figure 9: ASR (left) and $CLIP_p$ (right) values with varying further fine-tuning iterations.**

## 5.7 Backdoor Robustness

In real scenarios, computational overhead often leads users to download a publicly available pre-trained T2I diffusion model and fine-tune it with a small amount of their own customized data before deployment. Concurrently, fine-tuning is a prevalent method for mitigating backdoor attacks, where a defender uses clean training data to fin-tune a suspicious model, thereby eliminating any potential backdoors. We carry out experiments to investigate the robustness of EvilEdit after further fine-tuning. Specifically, we employ two different fine-tuning methods, full-parameter fine-tuning and LoRA [16] fine-tuning, on the pokemon-blip-captions [29] dataset. Both methods follow the default setting in Diffusers[1]. Fig. 9 illustrates that even after 1,500 steps of fine-tuning, the backdoor can still be activated with a high ASR (up to 80%). Overall, the results indicate that our backdoor is robust to further fine-tuning.

## 6 Conclusion

In this paper, we introduce EvilEdit, a novel approach for injecting backdoors into T2I diffusion models by directly editing the model parameters. EvilEdit reframes the backdoor injection as a model editing problem, aligning the trigger with the backdoor target by editing the projection matrices in the cross-attention layers of T2I models. Extensive experiment results demonstrate that EvilEdit surpasses existing backdoor attack methods in terms of practicality, effectiveness, and efficiency. Our work exposes significant vulnerabilities in current T2I diffusion models, laying the groundwork for future research into more advanced defense mechanisms.

---

[1]https://huggingface.co/docs/diffusers/training/text2image

## Acknowledgments

This work is supported by the National Key R&D Program of China under Grant 2022YFB3103500, the National Natural Science Foundation of China under Grant 62302071, the Natural Science Foundation of Chongqing, China, under Grants cstc2022ycjh-bgzxm0031, cstc2021jcyj-msxmX0744 and CSTB2023NSCQ-MSX0693, the National Research Foundation, Singapore and the Cyber Security Agency under its National Cybersecurity R&D Programme (NCRP25-P04-TAICeN), and the Singapore Ministry of Education (MOE) AcRF Tier 2 under Grant MOE-T2EP20121-0006.

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
