# OpenReview forum: "EvilEdit: Backdooring Text-to-Image Diffusion Models in One Second"
_acmmm.org/ACMMM/2024/Conference — MM2024 Poster_

### Official Review · Reviewer_SYsb · 2024-05-24

**Rating:** 4
**Confidence:** 2

**Summary:**

The paper introduces a novel method named “EvilEdit” for injecting backdoors into text-to-image (T2I) diffusion models. The method operates by directly editing the projection matrices within the cross-attention layers of the models, thereby aligning a chosen trigger with a specified backdoor target. This alignment ensures that when the trigger is present in the input, the model generates an image corresponding to the backdoor target, rather than the expected output. The paper also presents EvilEditVTA, an extension that allows using specific images as backdoor targets. The authors claim that their method can successfully backdoor a model within one second, with up to a 100% success rate, while only modifying a small fraction of the model's parameters and maintaining performance on benign prompts.

**Strengths:**

1. The novel backdoor attack method mentioned is not data poisoning-based, which means training- and data-free and also timesaving.
2. The paper demonstrates the effectiveness of EvilEdit in injecting backdoors into diffusion models, providing detailed method descriptions and experimental results.
3. The theoretical and empirical analyses support the high success rate of the backdoor attack, showcasing the alignment of trigger and backdoor target projections in cross-attention layers.

**Limitations:**

1. The paper primarily evaluates the method on the Stable Diffusion model. To further validate its robustness and generalizability, the method should be tested across a broader range of T2I diffusion models.
2. The paper mainly uses target attack, which lacks in-depth comparisons with other backdoor attack methods against Text-to-Image diffusion models, which could provide a broader perspective on the proposed approach.
3. In Section 4.3, when introducing the protected whitelist, it indicates that “when ’beautiful cat’ is used as the trigger, the protected whitelist should be {beautiful,cat}”,but you don’t mention the situation when the trigger contains multiple words and a part of them appears as a whole.
4. There are many places in the paper saying that you use “’beautiful cat’ as the trigger and ‘zebra’ as the backdoor target.”, however regardless of Figure 1 or any other places, I don’t see any visual representation about it.

**Suitability:**

2

---

### Official Review · Reviewer_4QSK · 2024-05-24

**Rating:** 4
**Confidence:** 2

**Summary:**

This paper presents a backdoor method on T2I diffusion models that edits the projection matrix.

**Strengths:**

1. The paper idea is novel by solving the optimization objective to get the analytical solution. Hence, the method does not need data and additional training to modify the model.
2. The evaluation results demonstrate its effectiveness in attack success rate, generation performance, and efficiency in terms of runtime, parameters, and samples required.

**Limitations:**

1. The author only qualitatively evaluates effects of protected whitelist. A quantitative comparison is needed to justify the claim that protected whitelist is in general effective.
2. Evaluations on text-to-image similarity (e.g. BLIP) or vision QA (e.g. BLIP-VQA) can be added. Currently, only CLIP model is used. As model-based evaluation tend to be biased, authors should evaluate using different models.
3. Evaluations only done on Stable Diffusion v1.5. How is the performance on other models. SD-XL applies a different architecture than SD1 and 2 models. It would be good to see the backdoor performance on this larger model.

**Suitability:**

3

---

### Official Review · Reviewer_n1oY · 2024-05-25

**Rating:** 5
**Confidence:** 3

**Summary:**

The paper presents an approach to inject backdoors into Text-to-Image (T2I) diffusion models, implementation of the attack does not require any training or additional data like traditional methods, it directly edits the projection matrices in the cross-attention layers instead, which is claimed to be training-Free and data-free.

**Strengths:**

- Implementation of the attack does not require  training or additional data like traditional methods, it directly edits the projection matrices in the cross-attention layers
- The paper provides a framework for backdoor injection as a model editing problem, both projection alignment and protected whitelists are clearly demonstrated and formulated.

**Limitations:**

- The proposed framework is claimed to be designed for T2I models, while experiments only covers StableDiffusion, results regarding how the method works on other types of commonly used T2I models is expected to be included
- Potential countermeasures are briefly mentions, but are not explored in detail, a thorough evaluation of existing defense mechanisms would be appreciated

**Suitability:**

3

---

### Official Review · Reviewer_3dX8 · 2024-05-26

**Rating:** 3
**Confidence:** 3

**Summary:**

To address the requirement of high computational overhead and substantial tuning data of backdooring text-to-image diffusion models, this paper proposing EvilEdit, which modifies the projection matrices to align the projection between triggers and the backdoor target using model editing techniques. Experimental results show its improvements of attack speed and data-free character.

**Strengths:**

Good motivation and novel idea: This method leverages model editing to efficiently inject backdoors under the training fine-tuning paradigm, which I believe is a promising direction. The experimental results demonstrate the satisfactory performance of this method.

**Limitations:**

Overall, this paper presents an interesting topic with a novel idea. However, the issues raised, mainly the Weakness 1, make me concern whether the experiments are solid enough to support the claims. I look forward to the authors' responses and would consider revising my score based on a satisfactory explanation.

Weakness 1. Counterintuitive and unexplained experimental results. One obvious true assumption for a backdoored model is that fine-tuning it on a clean dataset should not enhance the model's backdoor effectiveness. However, in Fig. 9, there are multiple data points that contradict this intuition: (1) For “Full-parameter” finetuning, the ASR at 15K steps (~90%) is significantly higher than at 14K steps (~72%). (2) For “LoRA” finetuning, at 6K steps is higher than at 3K, 4K, 5K steps.
The authors do not provide an explanation for these anomalies. One possible explanation for me could be an error in the ASR calculation. However, such significant discrepancies might make experimental evaluation not convincing.

Some small questions:
Q 1. Regarding the visual target backdoor, the authors use textual inversion to make the model learn the "target image." However, textual inversion cannot capture detailed object specifics [1]. Considering recent backdoor methods on text-to-image diffusion models [2, 3], which can enforce the model to generate pixel-level specific backdoors, have the authors considered the differences between their method and the baselines in this aspect?
Q 2. The authors use a whitelist to prevent the model from showing backdoor trigger effects on single words, but they only consider cases where the trigger consists of two tokens. If the trigger is tokenized into multiple word pieces, could the current method cause the backdoor to trigger with any combination of these word pieces?


[1] Gal, Rinon, et al. "An Image is Worth One Word: Personalizing Text-to-Image Generation using Textual Inversion." The Eleventh International Conference on Learning Representations. 2022.
[2] Zhai, Shengfang, et al. "Text-to-image diffusion models can be easily backdoored through multimodal data poisoning." Proceedings of the 31st ACM International Conference on Multimedia. 2023.
[3] Huang, Yihao, et al. "Personalization as a Shortcut for Few-Shot Backdoor Attack against Text-to-Image Diffusion Models." Proceedings of the AAAI Conference on Artificial Intelligence. Vol. 38. No. 19. 2024.

**Suitability:**

2

---

### Meta-Review · Area_Chair_5BMZ · 2024-07-01

**Recommendation:** Accept (Poster)
**Confidence:** 5

**Metareview:**

This paper proposed an approach to inject backdoors into Text-to-Image diffusion models. The paper received all acceptance after rebuttal. AC agreed with reviewers and voted for acceptance of this paper.